# A Prognostic Model Based on Nutritional Risk Index in Operative Breast Cancer

**DOI:** 10.3390/nu14183783

**Published:** 2022-09-14

**Authors:** Fei Lin, Wen Xia, Miao Chen, Tongchao Jiang, Jia Guo, Yi Ouyang, Haohui Sun, Xiaoyu Chen, Wuguo Deng, Ling Guo, Huanxin Lin

**Affiliations:** The State Key Laboratory of Oncology in South China, Collaborative Innovation Center for Cancer Medicine, Sun Yat-sen University Cancer Center, Guangzhou 510060, China

**Keywords:** breast cancer, nutritional risk index (NRI), prognosis, nomogram

## Abstract

Background: The nutritional risk index (NRI) is an independent prognostic factor for overall survival in various cancers, but its prognostic value in breast cancer remains unclear. This study aimed to explore the relationship between the NRI and overall survival (OS) in breast cancer and to develop a predictive nomogram. Methods: We retrospectively enrolled 1347 breast cancer patients who underwent mastectomy or lumpectomy between January 2011 and November 2012. Using a cutoff value of 110.59, patients were divided into a high-NRI group and a low-NRI group. OS was compared between the two groups. Clinicopathological factors independently associated with survival were used to construct a predictive nomogram. Results: Of the 1347 patients, 534 patients were classified as high NRI and 813 as low NRI. OS was significantly shorter in low-NRI patients. The 3- and 5-year OS rates were 87.3% and 73.4%, respectively, in the high-NRI group whereas they were 83.0% and 67.2%, respectively, in the low-NRI group. Cox regression analysis found that histopathological type, tumor size, lymph node status, progesterone receptor (PR) status, Ki-67, and NRI were independently associated with OS. Conclusions: NRI is an independent prognostic factor of OS in breast cancer patients. The proposed nomogram model may be a useful tool for individualized survival prediction.

## 1. Introduction

Breast cancer is the most common malignant tumor in women [1,2]. Although its mortality has decreased by more than 2% in some high-income countries [3], the latest epidemiological data for 2022 show that breast cancer remains to have the highest incidence in women, in both developing and developed countries [2]. Currently, although comprehensive treatment has a clear effect, early diagnosis and individualized precision treatment are still crucial, as local recurrence and distant metastasis remain common in breast cancer [4,5]. Furthermore, breast cancer patients at the same stage and receiving the same treatment may have completely different outcomes. In light of such prognostic heterogeneity [6], individualized precision treatment is key to treatment success. Thus, it is imperative that alternative biomarkers should be identified in order to improve prognostic stratification and prediction of treatment outcomes.

So far, many studies have indicated that cancer progression and metastasis depend not only on the type of tumor, method of treatment, stage of disease, and so on [7], but also on the patient’s nutritional status [8,9]. There is a complex interaction mechanism between nutrition and tumorigenesis [10]. In the past, body mass index (BMI) and serum albumin (ALB) were often used to assess the nutritional status of cancer patients. However, as a measure of nutritional status, BMI and albumin have various deficiencies. First, the proportion of body fat increases with age, whereas muscle mass decreases, but the corresponding changes in height, weight, and BMI may not reflect these changes, and so the specificity of BMI is poor [11,12]. Second, the level and function of albumin are affected by many liver-related diseases, such as liver cirrhosis, and patients with liver cirrhosis present post-translational modifications to albumin that compromise its level and function [13].

Based on the above, the evaluation of nutritional status by traditional parameters, such as BMI or albumin, inevitably limits the prediction power of the nutritional status of the patients. The nutritional risk index (NRI), calculated using the patient’s height, weight, and serum albumin level, has proven to be a more reliable evaluation of nutrition status than BMI and serum albumin in many types of cancer [14,15,16,17,18,19,20]. However, the prognostic value of NRI has rarely been reported in breast cancer. In this study, we built a novel prognostic model based on NRI and clinicopathological characteristics to predict overall survival in breast cancer patients. We retrospectively enrolled 1347 breast cancer patients who underwent mastectomy or lumpectomy and calculated the NRI for each patient. We compared the overall survival between the high-NRI and low-NRI groups, and identified the prognostic significance of NRI in breast cancer. We also constructed a predictive nomogram based on the NRI and clinicopathological factors. Our results demonstrated that NRI is an independent prognostic factor of overall survival in breast cancer patients, and the proposed nomogram model may be a useful tool for individualized survival prediction.

## 2. Materials and Methods

### 2.1. Patients

A total of 1347 patients diagnosed with breast cancer who underwent surgery at the Sun Yat-sen University Cancer Center (SYSUCC; Guangzhou, China) between January 2011 and November 2012 were included in this study. The inclusion criteria were: (1) breast cancer that was pathologically confirmed; (2) a mastectomy or lumpectomy had been performed. The exclusion criteria were: (1) breast cancer in men; (2) relapse and de novo; (3) ductal carcinoma in situ (DCIS); (4) the primary tumor was complicated by another primary tumor; (5) patients received any antitumor treatment before surgery; (6) the NRI indicator could not be calculated due to incomplete laboratory data; (7) inflammatory diseases such as hematological, autoimmune, or chronic/acute inflammation; (8) lost to follow-up. This study was approved by SYSUCC’s Research Ethics Committee (approval number: B2022-332-01). Patients’ names and IDs were kept confidential.

### 2.2. Data Collection and Classification

A list of patients who visited SYSUCC was obtained from the follow-up department. Laboratory data regarding the first admission of patients were searched using a case system and documented in detail using Microsoft Excel (Redmond, WA, USA). Clinical and laboratory data, such as patients’ height, weight, and serum albumin level, were gathered from the patients’ medical records one week prior to surgery. Indicators were calculated according to the following formulas: NRI = 1.487 × serum albumin concentration (g/L) + 41.7 × preoperative weight/ideal body weight (kg). Ideal body weight was defined as 22 × height (m)^2^ [21]. BMI = weight (kg)/height (m)^2^ [22]. The calculated NRI, BMI, and related indicators were analyzed in Microsoft Excel.

Patients were staged according to the 8th Edition American Joint Committee on Cancer-Tumor, Node, and Metastases (AJCC-TNM) staging system [23]. The expression of estrogen receptor (ER) and progesterone receptor (PR) were scored using the St. Gallen criteria [24]. According to the American Society of Clinical Oncology and College of American Pathologists guidelines, the human epidermal growth factor receptor (HER-2) status was assessed [25,26] by immunohistochemistry (IHC) or fluorescence in situ hybridization (FISH) assay. HER-2 negative status was defined as IHC HER-2+/++, or the FISH result was negative, or the FISH test was not performed; HER-2 positive status was defined as IHC = 3+, or FISH-positive/chromogenic in situ hybridization-positive.

### 2.3. Follow-Up

Patients were followed up, either by phone or through an outpatient surveillance system, to determine their condition or the date of death if the patient had already died. The study’s main endpoint was OS, defined as the time between the first diagnosis and death. For patients who survived, the date of the last follow-up was considered as the study endpoint; for those who had died before the study ended, the date of death was considered as the study endpoint.

### 2.4. Statistical Analysis

We report the median and interquartile range (IQR) for continuous variables, and the frequency and percentage for categorical variables. An analysis of the association between NRI groups and other clinicopathological features was conducted with the chi-square test (categorical variables) and Mann–Whitney U test (continuous variables), respectively. An optimal cut-off point for continuous variables was determined by selected rank statistics. Kaplan–Meier survival curves were plotted, and significance was assessed based on the log-rank test. Univariate and multivariate Cox regression analyses were performed to identify the factors influencing OS. Prior to performing the multivariate analysis, the proportional hazards assumption test was performed using Schoenfeld residuals. All factors with *p*-value < 0.05 detected in univariate analyses were included in multivariate analysis. A nomogram was generated based on the results of multivariate cox regression analysis. The performance of the nomogram was assessed by the concordance index (C-Index). We also applied 1000 bootstrap samples and 10-fold cross-validation to avoid overfitting. SPSS 25.0 (IBM Corp., Armonk, NY, USA) and R software (http://www.R-project.org; accessed date: 12 September 2021, version 4.0.2, R Core Team, Vienna, Austria) were used for all statistical analyses. All statistical tests were two-sided, and statistical significance was defined as *p*-value < 0.05.

## 3. Results

### 3.1. The Optimal Cut-Off Value of NRI

Using selected rank statistics, the optimal cutoff value for the NRI was determined as 110.59 for the whole cohort (Appendix A).

### 3.2. Clinical Characteristics of Patients and Their Relationship with NRI 

The characteristics of the patients and the relationship between the NRI and various clinicopathological factors of the whole cohort are presented in Table 1. The median age of the patients was 47 years (IQR, 40–58). According to the 8th Edition AJCC-TNM classification system, 379 (28.2%) patients were diagnosed with stage I, 624 (46.3%) were diagnosed with stage II, and 344 (25.5%) were diagnosed with stage III breast cancer. While 1137 (84.4%) patients had invasive ductal carcinoma, 210 (15.6%) were in other pathological types. The median BMI of the patients was 22 (IQR, 19.1–23). Meanwhile, 937 (69.6%) patients were ER-positive, 829 (61.5%) were PR-positive, and 398 (29.5%) were HER-2-positive. A total of 868 (64.4%) patients were Ki-67 > 14%, 1094 (81.2%) patients received adjuvant chemotherapy, and 364 (27%) patients received radiotherapy. The median value of the pretreatment serum albumin was 43.7g/L (IQR, 35.2–51.8). The median height of patients was 158.5 cm (IQR, 148.3–164.1), and the average weight of the patients was 57.6 kg (IQR, 45.5–68.7).

In addition, analysis of the relationship between the NRI and various clinicopathological factors revealed a significant association between the NRI and age (*p* = 0.006). The entire cohort was randomly divided into training and validation sets (ratio: 7:3), which is shown in Table 2. In the training and validation sets, 576 (61.1%) and 235 (58.2%) patients were assigned to the low-NRI groups, respectively.

### 3.3. Prognostic Value of NRI for Overall Survival (OS)

Based on the optimal cut-off value of NRI, the whole cohort was divided into two groups: a low-NRI group (NRI ≤ 110.59) and a high-NRI group (NRI > 110.59). In Figure 1, the Kaplan–Meier curve shows significantly shorter OS in the low-NRI group than in the high-NRI group (HR: 0.684, 95% confidence interval (CI): 0.478–0.980, log-rank test, *p* = 0.037). The 3- and 5- year OS rates were 87.3% and 73.4%, respectively, in the high-NRI group, and were 83.0% and 67.2%, respectively, in the low-NRI group.

### 3.4. Univariate and Multivariate Cox Regression Analyses

In the univariate Cox regression analysis, age, histopathological type, tumor size, lymph node status, ER status, PR status, HER-2 status, Ki-67, adjuvant chemotherapy, radiotherapy, endocrine therapy, targeted therapy, and the NRI were included. The results showed that the histopathological type, tumor size, lymph node status, PR status, HER-2 status, Ki-67, adjuvant chemotherapy, radiotherapy, and NRI were all significantly associated with OS. In the multivariate Cox regression analysis, only histopathological type, tumor size, lymph node status, PR status, Ki-67, and the NRI were independently associated with OS. The results of the univariate and multivariate Cox regression analyses are presented in Table 3.

In both the training and validation sets, we conducted survival analyses as well as univariate and multivariate Cox regression analyses (Appendix A). Appendix A present the Kaplan–Meier survival curves for the training and validation cohorts (training cohort: HR: 0.713, 95% CI: 0.547–0.892, log-rank test, *p* = 0.033; validation cohort: HR: 0.821, 95% CI: 0.599–1.328, log-rank test, *p* = 0.062). Although the *p*-value was not statistically significant in the validation cohort, the result was consistent with the whole cohort, the possible reason being the small sample size. Additionally, the results of the univariate and multivariate Cox regression analyses were in agreement for the whole cohort.

Using these independent factors, a prognostic model was established and graphically displayed as a nomogram to predict the 1-, 3-, and 5-year OS (Figure 2). With a C-Index of 0.793 (95% CI: 0.728–0.821), the prediction model showed suitable discriminating ability. In Figure 3, the calibration curves for the 1-, 3-, and 5-year OS illustrate good fitness between predicted and actual survival rates.

### 3.5. Subgroup Analysis of Common Clinical Variables

As shown in Figure 4, the subgroup analysis of common clinical variables showed that there was no correlation between the NRI and clinicopathological characteristics (all *p* > 0.05).

## 4. Discussion

In recent years, with the improvements in public health awareness and medical progression, the detection rate of early breast cancer has significantly increased, and the mortality rate is decreasing [27]. Further analysis of the cause of death revealed that approximately 20–30% of cancer-related deaths are due to malnutrition and cachexia, rather than the cancer itself [28,29]. Malnutrition adversely affects the body’s immune system and treatment efficacy, and then accelerates disease progression, local recurrence, and distant metastasis [30,31,32]. Thus, nutritional support treatment in combination with antitumor therapy is indispensable.

The American Joint Committee on Cancer (AJCC) TNM staging system is used as the benchmark by which to predict the prognosis and guide the treatment of breast cancer [33]. Higher stages indicate worse prognosis, and lower stages indicate better prognosis. Previous versions of the TNM staging system were mainly based on anatomical factors, the number of metastatic lymph nodes, and the presence of distant metastasis. It does not comprehensively consider the biological heterogeneity of patients or other risk factors [33,34], such as molecular subtypes [35], family history, germline mutations in driver genes (such as BRCA1) [36], etc. Such issues might affect the prediction accuracy of the conventional system [37,38]. In November 2016, AJCC promulgated the 8th Edition of the breast cancer staging system, which was globally implemented on 1 January 2018. Compared with previous versions, the 8th Edition of the AJCC cancer staging system for breast cancer proposed the concept of a prognostic stage group. Prognostic staging includes anatomical TNM staging, tumor histological grade (G), expression status of biomarkers (ER, PR, and HER-2), and a multigene staging system for detecting the risk of recurrence. Overall, the transition to the 8th Edition is expected to improve clinical care, treatment recommendations, and future research [39]. The TNM staging system is used to facilitate the description and classification of malignant tumors in the body. In this study, we proved that the NRI, a new nutrition-related indicator, has value in predicting the prognosis of breast cancer patients. From the perspective of nutrition, we compared the NRI with the traditional nutritional evaluation indicators: BMI and ALB. The time-dependent ROC curves show that NRI is more accurate in predicting OS than BMI and ALB (Appendix A). A recent study concluded that the prognostic and predictive performance of the NRI was superior to that of BMI and ALB [40].

As to the results of survival analyses and univariate and multivariate Cox regression analyses, in both the training and validation sets, although the *p*-value was not statistically significant in the validation cohort, the result was consistent with that of the whole cohort. A possible reason is the small sample size. Additionally, the results of the univariate and multivariate Cox regression analyses agreed for the whole cohort.

The NRI and age were significantly associated, according to the correlation analysis in our study, but the exact reason for this association is unclear. Selection bias is one of the possible causes. Despite the statistical significance of this association, it remains to be examined whether there is a true clinical significance in future studies.

Based on subgroup analysis, we found no association between the NRI and clinicopathological characteristics in our study, though we statistically proved that the NRI had predictive consistency within each subgroup. The clinical value of the NRI should be further confirmed by larger-scale data and prospective studies in the future.

A nomogram incorporating several prognostic factors can be a convenient tool in clinical settings, individualized predictions of survival, and deciding the optimal interval for follow-up and imaging [41,42]. Our calibration curves based on the nomogram performed well, which indicated that the nomogram we built by incorporating six independent prognostic factors is valuable. These six indicators were based on the results of our multifactor analysis. The main strength of this study is that we offered evidence that the NRI is independently related to survival outcomes in patients with breast cancer. We hope that more studies can validate and confirm the application of the NRI in other cancers in the future.

NRI has been previously shown to be an independent prognostic factor in several other cancer types, including colorectal, non-small-cell lung, and bladder cancer [43,44,45]. We showed that NRI is an independent prognostic factor in breast cancer. One study classified elderly colorectal cancer patients into four groups based on the NRI and found that the worst prognosis was in the lowest NRI group and the best prognosis was in the highest NRI group [46]. In another study, patients with malnutrition were significantly older than those without malnutrition [47]. This conclusion reiterates that we must pay special attention to the nutritional status of older patients. So far, only a few studies have been conducted on the NRI in breast cancer. A previous study [48] showed that patients with high NRI values tended to have longer DFS and OS than those with low NRI values, in a total of 785 breast cancer patients. Similar conclusions were reached in the above-mentioned study; however, compared with our cut-off value, this study had a different conclusion. We classified the candidate continuous index according to the cut-off point determined by the maximally selected rank statistics using the “maxstat” package of R software [49], a widely recognized and applied method in many studies [50,51,52]. In the study mentioned above, they used the ROC curve to obtain the cut-off value. Therefore, the cut-off value of NRI varies between studies, which limits the clinical use of this biomarker.

Our current study has some limitations. First, this was a retrospective single-center study with a relatively small sample size. Moreover, because all patients were Chinese, the results are more applicable to Asian populations. Second, we only considered the preoperative nutritional status and did not assess the nutritional changes during and after treatment, which could have changed the outcome. Third, many predictive biomarkers have been identified, and some studies have already demonstrated the predictive value of the indicators, such as the Glasgow Prognostic Score (GPS) [53,54], Controlling Nutritional Status (CONUT) [55,56], Prognostic Nutrition Index (PNI) [57], and so on.

This study has the potential to support the current understanding that the NRI is independently associated with OS in patients with breast cancer. We hope that more applications of the NRI to breast cancer will be confirmed in the future.

## 5. Conclusions

The NRI is an independent predictor of overall survival (OS) in patients with breast cancer treated with surgery. The proposed nomogram model may be a useful tool for individualized assessment of the prognosis.

## Figures and Tables

**Figure 1 nutrients-14-03783-f001:**
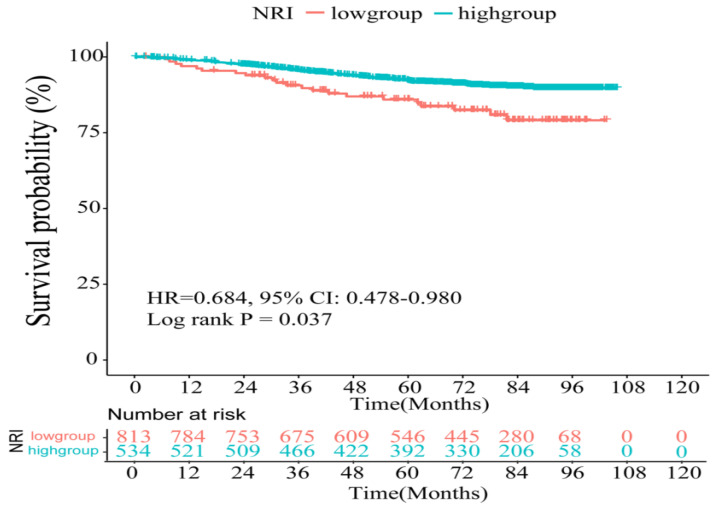
Kaplan–Meier survival curve of breast cancer patients after surgery in the whole cohort.

**Figure 2 nutrients-14-03783-f002:**
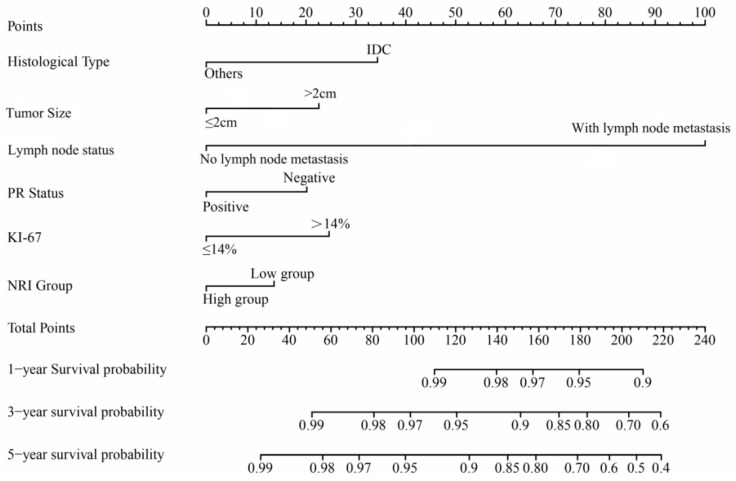
A nomogram for predicting the 1-, 3-, and 5-year overall survival in breast cancer patients using the whole cohort. Patients were staged according to the 8th Edition American Joint Committee on Cancer-Tumor, Node and Metastases staging system. PR: progesterone receptor; NRI Group: low group (≤110.59) and high group (>110.59).

**Figure 3 nutrients-14-03783-f003:**
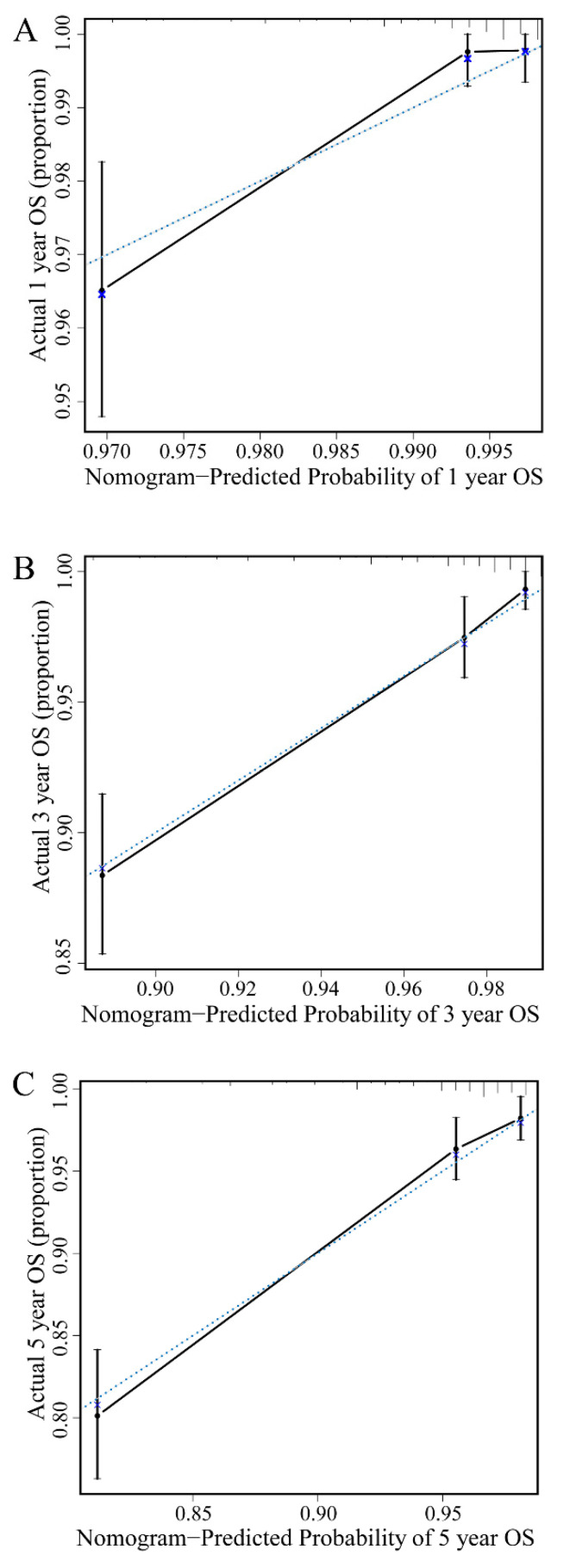
Nomogram model calibration plot for 1-year (**A**), 3-year (**B**), and 5-year (**C**) overall survival in the whole cohort.

**Figure 4 nutrients-14-03783-f004:**
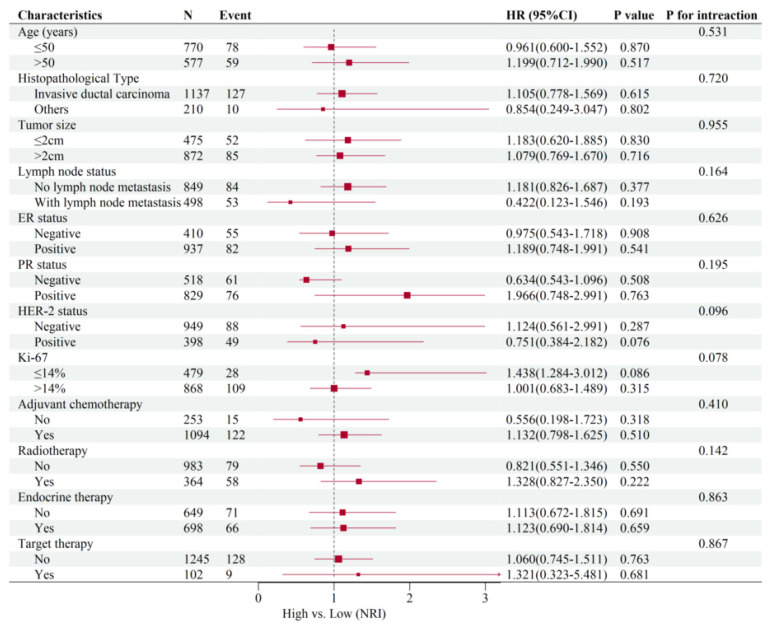
Forest plot of the results of multivariable regression analysis.

**Table 1 nutrients-14-03783-t001:** The relationship between the NRI and clinicopathological characteristics in patients in the whole cohort (*n* = 1347).

Characteristic Total No. (%) (*n* = 1347)		Low-NRI Group (*n* = 813)	High-NRI Group (*n* = 534)	*p*
Age (years) median (IQR)	47 (40–58)	48 (41–59)	47(40–59)	0.006
Histological type				
Invasive ductal carcinoma	1137 (84.4%)	687 (51.0%)	450 (33.4%)	0.908
Others	210 (15.6%)	126 (9.4%)	84 (6.2%)	
Tumor size				
≤2 cm	475 (35.3%)	300 (22.3%)	175 (13.0%)	0.624
>2 cm	872 (64.7%)	513 (38.1%)	359 (26.6%)	
Lymph node status				
No lymph node metastasis	849 (63.1%)	487 (36.2%)	362 (26.9%)	0.810
With lymph node metastasis	498 (36.9%)	294 (21.8%)	204 (15.1%)	
Clinical stage				
I	379 (28.2%)	253 (18.8%)	126 (9.4%)	0.753
II	624 (46.3%)	315 (23.4%)	309 (22.9%)	
III	344 (25.5%)	245 (18.2%)	99 (7.3%)	
BMI kg/m^2^, median (IQR)	22 (19.1–23)	21 (19.7–22.0)	21 (20.0–22.0)	0.189
ER status				
Positive	937 (69.6%)	561 (41.7%)	376 (27.9%)	0.583
Negative	410 (30.4%)	252 (18.7%)	158 (11.7%)	
PR status				
Positive	829 (61.5%)	505 (37.5%)	324 (24.0%)	0.595
Negative	518 (38.5%)	308 (22.9%)	210 (15.6%)	
HER-2 status				
Positive	398(29.5%)	234 (17.4%)	164 (12.1%)	0.357
Negative	949 (70.5%)	580 (43.1%)	369 (27.4%)	
Ki-67				
>14%	868 (64.4%)	499 (37.0%)	369 (27.4%)	0.300
≤14%	479 (35.6%)	314 (23.3%)	165 (12.3%)	
Type of Surgery				
Modified radical mastectomy	1078 (80.0%)	645 (47.9%)	433 (32.1%)	0.432
Others	269 (20.0%)	168 (12.5%)	101 (7.5%)	
Radiotherapy				
Yes	364 (27.0%)	218 (16.2%)	146 (10.8%)	0.831
No	983 (73.0%)	595 (44.2%)	388 (28.8%)	
Adjuvant chemotherapy				
Yes	1094 (81.2%)	654 (48.5%)	440 (32.7%)	0.369
No	253 (18.8%)	159 (11.8%)	94 (7.0%)	
Endocrine therapy				
Yes	698 (51.9%)	421 (31.3%)	277 (20.6%)	0.955
No	649 (48.1%)	392 (29.1%)	257 (19.0%)	
Target therapy				
Yes	102 (7.6%)	78 (5.8%)	24 (1.8%)	0.561
No	1245 (92.4%)	590(43.8%)	655 (48.6%)	

Abbreviations: NRI, Nutritional Risk Index; IQR, Inter-Quarter Range; BMI, Body Mass Index; ER: Estrogen Receptor; PR: Progesterone Receptor; HER2: Human Epidermal Growth Factor Receptor-2. Patients were staged according to the 8th American Joint Committee on Cancer-Tumor, Node, and Metastases (AJCC-TNM) stage.

**Table 2 nutrients-14-03783-t002:** The baseline haracteristics Betbween the training and validation datasets.

Characteristics	Training Set (*n* = 943)	Validation Set (*n* = 404)
Age (Years) Median (IQR)	47 (41–56)	48 (42–57)
Tumor size		
≤2 cm	340 (36.0%)	145 (35.9%)
>2 cm	603 (63.9%)	259 (64.1%)
Lymph node status		
No lymph node metastasis	481 (51.0%)	210 (52.0%)
With lymph node metastasis	462 (49.0%)	194 (48.0%)
Clinical stage		
I	210 (22.2%)	86 (21.2%)
II	516 (54.7%)	223 (55.3%)
III	217 (23.1%)	95 (23.5%)
Histological type		
Invasive ductal carcinoma	760 (80.6%)	316 (78.2%)
Others	183 (19.4%)	88 (21.8%)
ER status		
Positive	658 (69.8%)	284 (70.3%)
Negative	285 (30.2%)	120 (29.7%)
PR status		
Positive	636 (67.4%)	272 (67.3%)
Negative	307 (32.6%)	132 (32.7%)
HER-2 status		
Positive	249 (26.4%)	75 (18.6%)
Negative	694 (73.6%)	329 (81.4%)
Ki-67		
>14%	386 (41.0%)	151 (37.4%)
≤14%	557 (59.0%)	253 (62.6%)
NRI		
>110.59	367 (38.9%)	169 (41.8%)
≤110.59	576 (61.1%)	235 (58.2%)

Abbreviations: NRI, nutritional risk index; IQR, interquartile range; ER: estrogen receptor; PR: progesterone receptor; HER2: human epidermal growth factor receptor-2. Patients were staged according to the 8th American Joint Committee on Cancer-Tumor, Node, and Metastases (AJCC-TNM).

**Table 3 nutrients-14-03783-t003:** Univariate analysis and multivariate regression analyses of overall survival by Cox regression analysis.

Characteristics	Univariate Analysis Hazard Ratio (95% CI)		Multivariate Analysis Hazard Ratio (95% CI)	
		*p*		*p*
Age (years)				
≤50	1		-	-
>50	1.134 (0.810–1.587)	0.465	-	-
Histopathological Type				
Invasive ductal carcinoma	1		1	
Others	0.422 (0.220–0.800)	0.009 *	0.414 (0.182–0.775)	0.006 *
Tumor size				
≤2 cm	1		1	
>2 cm	2.419 (1.551–3.788)	<0.001 *	2.576 (1.231–2.742)	0.035 *
Lymph node status				
No lymph node metastasis	1		1	
With lymph node metastasis	5.527 (3.803–8.060)	<0.001 *	5.102 (3.598–6.350)	<0.001 *
ER status				
Negative	1		-	-
Positive	0.603 (0.426–1.844)	0.600	-	-
PR status				
Negative	1		1	
Positive	0.547 (0.391–0.764)	<0.001 *	0.687 (0.398–0.812)	0.009 *
HER-2 status				
Negative	1		1	
Positive	1.717 (1.219–2.419)	0.002 *	1.230 (0.687–1.701)	0.463
Ki-67				
≤14%	1		1	
>14%	2.197 (1.451–3.329)	<0.001 *	1.820 (1.231–2.664)	0.014 *
Adjuvant chemotherapy				
No	1		1	
Yes	1.770 (1.035–3.027)	0.037 *	2.386(0.552–2.798)	0.082
Radiotherapy				
No	1		1	
Yes	1.900 (1.355–2.669)	<0.001 *	1.701 (0.582–1.909)	0.148
Endocrine therapy				
No	1		-	-
Yes	0.762 (0.545–1.064)	0.111	-	-
Target therapy				
No	1		-	-
Yes	1.015 (0.497–2.073)	0.968	-	-
NRI				
≤110.59	1		1	
>110.59	0.684 (0.478–0.980)	0.037 *	0.620(0.505–0.890)	0.042 *

Abbreviations: NRI: nutritional risk index; ER: estrogen receptor; PR: progesterone receptor; HER2: human epidermal growth factor receptor-2. Patients were staged according to the 8th American Joint Committee on Cancer-Tumor, Node, and Metastases (AJCC-TNM). The symbol “*” was indicate that the *p* value is less than 0.05.

## Data Availability

The data presented in this study are available on request from the corresponding author. The data are not publicly available due to hospital copyright.

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
