# Peer review of "A Prognostic Model Based on Nutritional Risk Index in Operative Breast Cancer"

_nutrients, 2022, doi:10.3390/nu14183783_

Round 1
Reviewer 1 Report
This retrospective study is well designed and the methodology of the study and results are clearly explained. It is very commendable that there is a NRI as part of the life interventions in the process of breast cancer treatment. Its cut-off values and thus the nomogram should certainly be optimized.
In the discussion section, I recommend moderating the comparison of NRI with the TNM system, since TNM is used to facilitate the description and classification of malignant tumors in the body. The sentence" In this study, we compared NRI and traditional TNM staging system to determine which delivers the most accurate prognosis prediction" needs to be revised. So for example, in some cases, the triple negative cancer itself plays a critical role in the prognosis of breast cancer. Consultation with a senologist is desirable.
Reviewer 2 Report
Summary
The manuscript submitted by Fei Lin et al. analyzes the role of nutritional risk index (NRI) in establishing a prognostic model for the evaluation of overall survival (OS) in breast cancer. To this end, the authors conducted a retrospective observational study using a a single institution database (Sun Yat-sen University Cancer Center, Guangzhou, China) comprising 1347 breast cancer patients who underwent mastectomy or lumpectomy between January 2011 and November 2012. Patients were divided into two cohorts: a high-NRI group and a low-NRI group. OS was compared between the two groups. Clinicopathological factors such as histopathological type, tumor size, lymph node status, Estrogen Receptor (ER) status, Progesterone Receptor (PR) status, the human epidermal growth factor receptor (HER-2) status, Ki-67%, or adjuvant therapy, independently associated with survival, were overviewed, in order to design a predictive nomogram and establish whether or not these factors are linked with NRI in calculating the OS. The authors concluded that NRI is an independent prognostic factor for OS in breast cancer patients.
Title and abstract: The title and abstract are consistent with the
contents of the text, although, considering the limited scientific value of the
study (retrospective study, relatively small sample size, single institution
study on asian population), I find the title a little enthusiastic and
I advise a revision.
Introduction: The authors summarized the current available information on this topic in a clear and concise manner. The aim of the study is well stated and sustained by scientific data.
Materials and methods: The methodology for patient inclusion and exclusion was presented clearly. The protocol regarding data collection, follow-up, determination of NRI and adjustment of cut-of value, clinical and pathological definitions for the other variables is well established and structured. The statistical analysis is described, however, it is unclear why both, Mann-Whitney U- test and Student T-test, were used as they start from different premises (the normality of the distribution of data). Maybe a more comprehensive and detailed justification of the statistic methods is needed in order to raise the clarity of judgement in choosing the data analysis tools. This would ease the readers understanding of the subject.
Results: The authors adequately presented their findings, although on page 7, paragraphs 190-194, I find the statement „Although the p-value was not statistically significant in the validation cohort, the result was consistent with the whole cohort, the possible reason is the small sample size. Additionally, the results of univariate analysis and multivariate Cox regression analysis were in agreement with the whole cohort” inappropriate and I suggest to relocate the additional explanations only in the discussion chapter.
Discussions: The results are discussed in relation to the evidence currently available in the literature. The limitations and strengths of the present study are adequately presented. On page 11, paragraph 248, the author refers to figures 5 and 6, which are completely missing from the manuscript. I also advise that statements like „which reached a similar conclusion to ours” or „the conclusion was consistent with ours” should be revised in a more elegant and impersonal manner.
Conclusions: The conclusions of the authors are appropriately cautious given the limitations of the study.
Lastly, the manuscript is presented in a well-structured manner. Most of the cited references are recent and relevant. The use of language is mostly sound. Given the type of the study I find the authors be numerous and I suggest an adjustment of this aspect.
Author Response
Response to Reviewer 1 Comments
Point 1 : “This retrospective study is well designed and the methodology of the study and results are clearly explained. It is very commendable that there is a NRI as part of the life interventions in the process of breast cancer treatment. Its cut-off values and thus the nomogram should certainly be optimized.
In the discussion section, I recommend moderating the comparison of NRI with the TNM system, since TNM is used to facilitate the description and classification of malignant tumors in the body. The sentence" In this study, we compared NRI and traditional TNM staging system to determine which delivers the most accurate prognosis prediction" needs to be revised. So for example, in some cases, the triple negative cancer itself plays a critical role in the prognosis of breast cancer. Consultation with a senologist is desirable.”
Response 1: We thank the reviewer for the nice comment and kind advice. According to the suggestion of the reviewer, we have moderated the comparison of NRI with the TNM staging system since TNM is used to facilitate the description and classification of malignant tumors in the body. We have also deleted the imprecise description and revised the sentence in the Discussion section of the revised manuscript (Page 11).
Response to Reviewer 2 Comments
Point 1: “Title and abstract: The title and abstract are consistent with the contents of the text, although, considering the limited scientific value of the study (retrospective study, relatively small sample size, single institution study on Asian population), I find the title a little enthusiastic and I advise a revision.”
Response 1: We thank the reviewer for the kind suggestion. We have revised the title using the new one “A prognostic model based on nutritional risk index in operative breast cancer” (Page 1).
Point 2: “Introduction: The authors summarized the current available information on this topic in a clear and concise manner. The aim of the study is well stated and sustained by scientific data.”
Response 2: Thanks the reviewer for the nice comments.
Point 3: “Materials and methods: The methodology for patient inclusion and exclusion was presented clearly. The protocol regarding data collection, follow-up, determination of NRI and adjustment of cut-of value, clinical and pathological definitions for the other variables is well established and structured. The statistical analysis is described, however, it is unclear why both, Mann-Whitney U- test and Student T-test, were used as they start from different premises (the normality of the distribution of data). Maybe a more comprehensive and detailed justification of the statistic methods is needed in order to raise the clarity of judgement in choosing the data analysis tools. This would ease the readers understanding of the subject.”
Response 3: We thank the reviewer for the kind suggestions. Because the data did not conform to the normal distribution after the normality test, the continuous variable in the text were described by IQR, and the Mann-Whitney U-test was carried out. Corrections have been made in the revised “Materials and Methods”. Thanks for your suggestion (Page 3).
Point 4: Results: The authors adequately presented their findings, although on page 7, paragraphs 190-194, I find the statement “Although the p-value was not statistically significant in the validation cohort, the result was consistent with the whole cohort, the possible reason is the small sample size. Additionally, the results of univariate analysis and multivariate Cox regression analysis were in agreement with the whole cohort” inappropriate and I suggest to relocate the additional explanations only in the discussion chapter.
Response 4: Thanks for the reviewer’s kind advice. We have corrected the inappropriate statement and relocated the additional explanations in the Discussion section of the revised manuscript (Page 11).
Point 5: Discussions: The results are discussed in relation to the evidence currently available in the literature. The limitations and strengths of the present study are adequately presented. On page 11, paragraph 248, the author refers to figures 5 and 6, which are completely missing from the manuscript. I also advise that statements like “which reached a similar conclusion to ours” or “the conclusion was consistent with ours” should be revised in a more elegant and impersonal manner.
Response 5: We thank the reviewer for the nice comments and found the typing errors. We have corrected the wrong typing “Figures 5 and 6” as the “Supplementary Figure 4 and 5” in the revised manuscript (Page 11). We have also re-uploaded the supplementary pictures in the “Supplementary Material” part. In addition, we have also revised some statements in a more elegant and impersonal manner (Page 11).
Point 6: Conclusions: The conclusions of the authors are appropriately cautious given the limitations of the study.
Response 6: Thanks the reviewer for the nice comments.
Point 7: Lastly, the manuscript is presented in a well-structured manner. Most of the cited references are recent and relevant. The use of language is mostly sound. Given the type of the study I find the authors be numerous and I suggest an adjustment of this aspect.
Response 7: We thank the reviewer for the nice comments. Considering that each author contributed more or less to this article, sorry we keep the original author list. But we agree and accept you kind suggestion and keep it in mind, and will pay attention to this issue in the future.

Round 2
Reviewer 2 Report
The authors have sufficiently improved the manuscript according to previous recommendations.